# Pretraining Representations for Data-Efficient Reinforcement Learning

**Max Schwarzer**[*,1,2]**, Nitarshan Rajkumar**[*,1,2]**, Michael Noukhovitch**[1,2]**, Ankesh Anand**[1,2]
**Laurent Charlin**[1,3,5]**, Devon Hjelm**[1,4]**, Philip Bachman**[4]**, Aaron Courville**[1,2,5]

[1]Mila, [2]Université de Montréal, [3]HEC Montréal, [4]Microsoft Research, [5]CIFAR

## Abstract

Data efficiency is a key challenge for deep reinforcement learning. We address this problem by using unlabeled data to pretrain an encoder which is then finetuned on a small amount of task-specific data. To encourage learning representations which capture diverse aspects of the underlying MDP, we employ a combination of latent dynamics modelling and unsupervised goal-conditioned RL. When limited to 100k steps of interaction on Atari games (equivalent to two hours of human experience), our approach significantly surpasses prior work combining offline representation pretraining with task-specific finetuning, and compares favourably with other pretraining methods that require orders of magnitude more data. Our approach shows particular promise when combined with larger models as well as more diverse, task-aligned observational data – approaching human-level performance and data-efficiency on Atari in our best setting. We provide code associated with this work at https://github.com/mila-iqia/SGI.

## 1 Introduction

Deep reinforcement learning (RL) methods often focus on training networks *tabula rasa* from random initializations without using any prior knowledge about the environment. In contrast, humans rely a great deal on visual and dynamics priors about the world to perform decision making efficiently (Dubey et al., 2018; Lake et al., 2017). Thus, it is not surprising that RL algorithms which learn *tabula rasa* suffer from severe overfitting (Zhang et al., 2018) and poor sample efficiency compared to humans (Tsividis et al., 2017).

Self-supervised learning (SSL) provides a promising approach to learning useful priors from past data or experiences. SSL methods leverage unlabelled data to learn strong representations, which can be used to bootstrap learning on downstream tasks. Pretraining with self-supervised learning has been shown to be quite successful in vision (Hénaff et al., 2019; Grill et al., 2020; Chen et al., 2020a) and language (Devlin et al., 2019; Brown et al., 2020) settings.

Pretraining can also be used in an RL context to learn priors over representations or dynamics. One approach to pretraining for RL is to train agents to explore their environment in an unsupervised fashion, forcing them to learn useful skills and representations (Hansen et al., 2020; Liu & Abbeel, 2021; Campos et al., 2021). Unfortunately, current unsupervised exploration methods require months or years of real-time experience, which may be impractical for real-world systems with limits and costs to interaction — agents cannot be run faster than real-time, may require significant human oversight for safety, and can be expensive to run in parallel. It is thus important to develop pretraining methods that work with practical quantities of data, and ideally that can be applied *offline* to fixed datasets collected from prior experiments or expert demonstrations (as in Stooke et al., 2021).

---

[*]{max.schwarzer, nitarshan}@mila.quebec

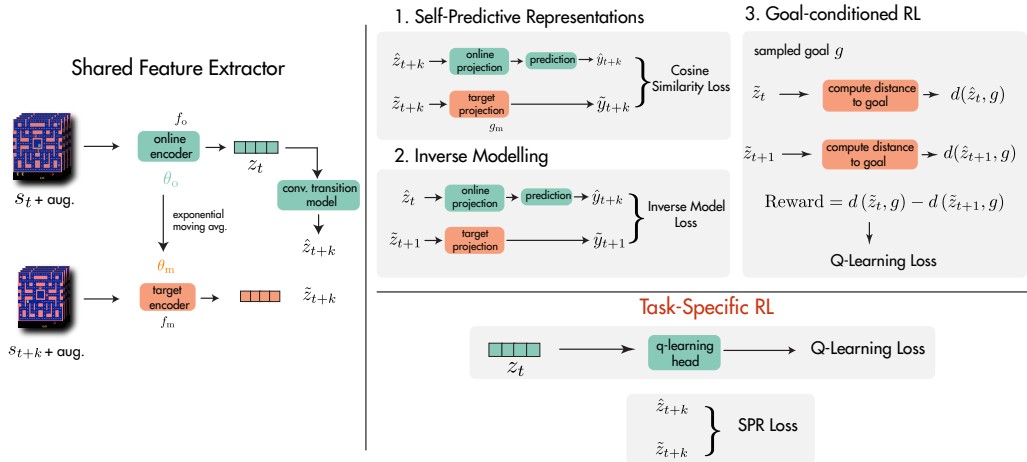

Figure 1: A schematic diagram showing our two stage pretrain-then-finetune method. All unsupervised training losses and task-specific RL use the shared torso on the left.

To this end, we propose to use a combination of self-supervised objectives for representation learning on offline data, requiring orders of magnitude less pretraining data than existing methods, while approaching human-level data-efficiency when finetuned on downstream tasks. We summarize our work below:

**RL-aligned representation learning objectives:** We propose to pretrain representations using a combination of latent dynamics modeling, unsupervised goal-conditioned reinforcement learning, and inverse dynamics modeling – with the intuition that a collection of such objectives can capture more information about the dynamical and temporal aspects of the environment of interest than any single objective. We refer to our method as **SGI** (**S**PR, **G**oal-conditioned RL and **I**nverse modeling).

**Significant advances for data-efficiency on Atari:** SGI's combination of objectives performs better than any in isolation and significantly improves performance over previous representation pretraining baselines such as ATC (Stooke et al., 2021). Our results are competitive with exploration-based approaches such as APT or VISR (Liu & Abbeel, 2021; Hansen et al., 2020), which require two to three orders of magnitude more pretraining data and the ability to interact with the environment during training, while SGI can learn with a small offline dataset of exploration data.

**Scaling with data quality and model size:** SGI's performance also scales with data quality and quantity, increasing as data comes from better-performing or more-exploratory policies. Additionally, we find that SGI's performance scales robustly with model size; while larger models are unstable or bring limited benefits in standard RL, SGI pretraining allows their finetuning performance to significantly exceed that of smaller networks.

We assume familiarity with RL in the following sections (with a brief overview in Appendix A).

## 2 Representation Learning Objectives

A wide range of SSL objectives have been proposed for RL which leverage various aspects of the structure available in agent interactions. For example, the temporal dynamics of an environment can be exploited to create a forward prediction task (e.g., Gelada et al., 2019; Guo et al., 2018; Stooke et al., 2021; Schwarzer et al., 2021) in which an agent is trained to predict its immediate future observations, perhaps conditioned on a sequence of actions to perform.

Structure in RL goes far beyond forward dynamics, however. We propose to combine multiple representation learning objectives, covering different agent-centric and temporal aspects of the MDP. Based on the intuition that knowledge of an environment is best represented in multiple ways (Hessel et al., 2021; Degris & Modayil, 2012), we expect this to outperform monolithic representation learning methods such as temporal contrastive learning (e.g., Stooke et al., 2021). In deciding which

tasks to use, we consider questions an adequate representation should be able to answer about its environment, including:

- If I take action $a$ in state $s$, what state $s'$ am I likely to see next?
- If I transitioned from state $s$ to state $s'$, what action $a$ did I take?
- What action $a$ would I take in state $s$ so that I reach another state $s'$ as soon as possible

Note that none of these questions are tied to task reward, allowing them to be answered in a fully-unsupervised fashion. Additionally, these are questions about the environment, and not any specific policy, allowing them to be used in offline pretraining with arbitrary behavioral policies.

In general, the first question may be answered by forward dynamics modeling, which as mentioned above is well-established in RL. The second question corresponds to inverse dynamics modeling, which has also been commonly used in the past (Lesort et al., 2018). The third question corresponds to self-supervised goal-conditioned reinforcement learning which has the advantage of being structurally similar to the downstream target task, as both require learning to control the environment.

To facilitate their joint use, we formulate these objectives so that they operate in the latent representation space provided by a shared encoder. We provide an overview of these components in Figure 1 and describe them in greater detail below; we also provide detailed pseudocode in Appendix D.

## 2.1 Self-Predictive Representations

SPR (Schwarzer et al., 2021) is a representation learning algorithm developed for data-efficient reinforcement learning. SPR learns a latent-space transition model, directly predicting representations of future states without reconstruction or negative samples. As in its base algorithm, Rainbow (Hessel et al., 2018), SPR learns a convolutional encoder, denoted as $f_o$, which produces representations of states as $z_t = f_o(s_t)$. SPR then uses a *dynamics model* $h$ to recursively estimate the representations of future states, as $\hat{z}_{t+k+1} = h(\hat{z}_{t+k}, a_{t+k})$, beginning from $\hat{z}_t \triangleq z_t$. These representations are projected to a lower-dimensional space by a projection function $p_o$ to produce $\hat{y}_{t+k} \triangleq p_o(\hat{z}_{t+k})$.

Simultaneously, SPR uses a *target encoder* $f_m$ to produce target representations $\tilde{z}_{t+k} \triangleq f_m(s_{t+k})$, which are further projected by a target projection function $p_m$ to produce $\tilde{y}_{t+k} \triangleq p_m(\tilde{z}_{t+k})$. SPR then maximizes the cosine similarity between these predictions and targets, using a learned linear prediction function $q$ to translate from $\hat{y}$ to $\tilde{y}$:

$$\mathcal{L}_\theta^{\text{SPR}}(s_{t:t+K}, a_{t:t+K}) = -\sum_{k=1}^{K} \frac{q(\hat{y}_{t+k}) \cdot \tilde{y}_{t+k}}{||q(\hat{y}_{t+k})||_2 \cdot ||\tilde{y}_{t+k}||_2}. \tag{1}$$

The parameters of these target modules $\theta_m$ are defined as an exponential moving average of the parameters $\theta_o$ of $f_o$ and $p_o$: $\theta_m = \tau\theta_m + (1-\tau)\theta_o$.

## 2.2 Goal-Conditioned Reinforcement Learning

Inspired by works such as Dabney et al. (2021) that show that modeling many different value functions is a useful representation learning objective, we propose to augment SPR with an unsupervised goal-conditioned reinforcement learning objective. We define goals $g$ to be normalized vectors of the same size as the output of the agent's convolutional encoder (3,136- or 4,704-dimensional vectors, for the architectures we consider). We use these goals to annotate transitions with synthetic rewards, and train a modified version of Rainbow (Hessel et al., 2018) to estimate $Q(s_t, a, g)$, the expected return from taking action $a$ in state $s_t$ to reach goal $g$ if optimal actions are taken in subsequent states.

We select goals using a scheme inspired by hindsight experience replay (Andrychowicz et al., 2017), seeking to generate goal vectors that are both semantically meaningful and highly diverse. As in hindsight experience replay, we begin by sampling a state from another trajectory or the future of the current trajectory. However, we take the additional step of applying stochastic noise to encourage goals to lie somewhat off of the current representation manifold. We provide details in Appendix C.2.

## 2.3 Inverse Dynamics Modeling

We propose to use an inverse dynamics modeling task (Lesort et al., 2018), in which the model is trained to predict $a_t$ from $s_t$ and $s_{t+1}$. Because this is a classification task (in discrete control) or

regression task (continuous control), it is naturally not prone to representational collapse, which may complement and stabilize our other objectives. We directly integrate inverse modeling into the rollout structure of SPR, modeling $p(a_{t+k}|\hat{y}_{t+k}, \tilde{y}_{t+k+1})$ for each $k \in (0, \ldots, K-1)$, using a two-layer MLP trained by cross-entropy.

## 3 Related Work

**Data-Efficiency**   In order to address data efficiency in RL, Kaiser et al. (2019) introduced the Atari 100k benchmark, in which agents are limited to 100,000 steps of environment interaction, and proposed SimPLe, a model-based algorithm that substantially outperformed previous model-free methods. However, van Hasselt et al. (2019) and Kielak (2020) found that simply modifying the hyperparameters of existing model-free algorithms allowed them to exceed SimPLe's performance. Later, DrQ (Kostrikov et al., 2021) found that adding mild image augmentation to model-free methods dramatically enhanced their sample efficiency, while SPR (Schwarzer et al., 2021) combined data augmentation with an auxiliary self-supervised learning objective. SGI employs SPR as one of its objectives in offline pretraining, leading to significant improvements in data-efficiency.

**Exploratory pretraining**   A number of recent works have sought to improve reinforcement learning via the addition of an unsupervised *pretraining* stage prior to finetuning on the target task. One common approach has been to allow the agent a period of fully-unsupervised interaction with the environment, during which the agent is trained to maximize a surrogate exploration-based task such as the diversity of the states it encounters, as in APT (Liu & Abbeel, 2021) or ProtoRL (Yarats et al., 2021), or to learn a set of skills associated with different paths through the environment, as in DIAYN (Eysenbach et al., 2018), VISR (Hansen et al., 2020), and DADS (Sharma et al., 2019). Others have proposed to use self-supervised objectives to generate intrinsic rewards encouraging agents to visit new states; e.g. Pathak et al. (2017) and Burda et al. (2018) use the loss of an inverse dynamics model like that used in SGI, while Sekar et al. (2020) uses the disagreement between an ensemble of latent-space dynamics models. Finally, Campos et al. (2021) report strong results based on massive-scale unsupervised pretraining.

Many of these methods are used to pretrain agents that are later adapted to specific reinforcement learning tasks. However, SGI differs in that it can be used offline and is agnostic to how data is collected. As such, if no pre-existing offline data is available, one of the methods above can be used to generate a dataset for SGI; we use Burda et al. (2018) for this in Section 4.2.

**Visual Representation Learning**   Computer vision has seen a series of dramatic advances in self-supervised representation learning, including contrastive methods (Oord et al., 2018; Hjelm et al., 2019; Bachman et al., 2019; He et al., 2020; Chen et al., 2020a) as well as purely predictive ones (Grill et al., 2020). Variants of these approaches have also been shown to improve performance when coupled with a small quantity of labeled data, in a *semi-supervised* setting (Chen et al., 2020b; Hénaff et al., 2019), and several self-supervised methods have been designed specifically for this case (for example, Sohn et al., 2020; Tarvainen & Valpola, 2017).

These advances have spurred similar growth in methods aimed specifically at improving performance in RL. We refer the reader to Lesort et al. (2018) for a review of earlier methods, including inverse dynamics modeling which is used in SGI. Recent research has focused on leveraging latent-space dynamics modeling as an auxiliary task. Gelada et al. (2019) propose a simple next-step prediction task, coupled with reward prediction, but found it is prone to latent space collapse and requires an auxiliary reconstruction loss for experiments on Atari. Guo et al. (2020) use a pair of networks for both forward and backward prediction, and show improved performance in extremely large-data multi-task settings. Mazoure et al. (2020) use a temporal contrastive objective for representation learning, and show improvement in continual RL settings. Concurrently, SPR (Schwarzer et al., 2021) proposed a multi-step latent prediction task with similarities to BYOL (Grill et al., 2020).

Two works similar to ours, Anand et al. (2019) and Stooke et al. (2021), propose reward-free temporal-contrastive methods to pretrain representations. Anand et al. (2019) show that representations from encoders trained with ST-DIM contain a great deal of information about environment states, but they do not examine whether or not representations learned via their method are, in fact, useful for reinforcement learning. However, Stooke et al. (2021) employ a similar algorithm and find only relatively minor improvements in performance compared to standard baselines in the large-data

regime; our controlled comparisons show that SGI's representations are far better for data-efficiency. Concurrent to our work, FERM (Zhan et al., 2020) propose contrastive pretraining from human demonstrations in a robotics setting. As FERM is quite similar to ATC, we are optimistic that our improvements over ATC in Atari 100k would translate to FERM's setting. Finally, Yang & Nachum (2021) propose a family of algorithms for learning representations for DRL from state observations; we believe that these might be promising in DRL from pixels, but they have not yet been adapted to that setting.

# 4    Experimental Details

In our experiments, We seek to address two main challenges for the deployment of RL agents in the real world (Dulac-Arnold et al., 2020): (1) training the RL agent with a limited budget of interactions in the real environment, and (2) leveraging existing interaction data of **arbitrary quality**.

## 4.1    Environment and Evaluation

To address the first challenge, we focus our experimentation on the Atari 100k benchmark introduced by Kaiser et al. (2019), in which agents are allowed only 100k steps of interaction with their environment.[2] This is roughly equivalent to the two hours human testers were given to learn these games by Mnih et al. (2015), providing a baseline of human sample-efficiency.

Atari is also an ideal setting due to its complex observational spaces and diverse tasks, with 26 different games included in the Atari 100k benchmark. These factors have led to Atari's extensive use for representation learning and exploratory pretraining (Anand et al., 2019; Stooke et al., 2021; Campos et al., 2021), and specifically Atari 100k for data-efficient RL (e.g., Kaiser et al., 2019; Kostrikov et al., 2021; Schwarzer et al., 2021), including finetuning after exploratory pretraining (e.g., Hansen et al., 2020; Liu & Abbeel, 2021), providing strong baselines to compare to.

Our evaluation metric for an agent on a game is *human-normalized score* (HNS), defined as $\frac{agent\_score - random\_score}{human\_score - random\_score}$. We calculate this per game by averaging scores over 100 evaluation trajectories at the end of training, and across 10 random seeds for training. We report both mean (Mn) and median (Mdn) HNS over the 26 Atari-100K games, as well as on how many games a method achieves super-human performance ($>$H) and greater than random performance ($>$0). Following the guidelines of Agarwal et al. (2021) we also report interquartile mean HNS (IQM) and quantify uncertainty via bootstrapping in Appendix B.

## 4.2    Pretraining Data

The second challenge pertains to pretraining data. Although some prior work on offline representational pretraining has focused on expert-quality data (Stooke et al., 2021), we expect real-world pretraining data to be of greatly varying quality. We thus construct four different pretraining datasets to approximate different data quality scenarios.

- **(R)andom** To assess performance near the lower limit of data quality, we use a random policy to gather a dataset of 6M transitions for each game. To encourage the agent to venture further from the starting state, we execute each action for a random number of steps sampled from a Geometric($\frac{1}{3}$) distribution.

- **(E)xploratory** To emulate slightly better data that covers a larger range of the state space, we use an exploratory policy. Specifically, we employ the **IDF** (inverse dynamics) variant of the algorithm proposed by Burda et al. (2018). We log the first 6M steps from an agent trained in each game. This algorithm achieves better-than-random performance on only 70% of tasks, setting it far below the performance of more modern unsupervised exploration methods.

To create higher-quality datasets, we follow Stooke et al. (2021) and use experience gathered during the training of standard DQN agents (Mnih et al., 2015). We opt to use the publicly-available DQN Replay dataset (Agarwal et al., 2020), which contains data from training for 50M steps (across all 57 games, with five different random seeds). Although we might prefer to use data from recent

---

[2]Note that sticky actions are disabled under this benchmark.

unsupervised exploration methods such as APT (Liu & Abbeel, 2021), VISR (Hansen et al., 2020), or CPT (Campos et al., 2021), none of these works provide code or datasets, making this impractical. We address using data collected from on-task agents with a behavioural cloning baseline in Section 5, with surprising findings relative to prior work.

- **(W)eak** We first generate a weak dataset by selecting the first 1M steps for each of the 5 available runs in the DQN Replay dataset. This data is generated with an $\epsilon$-greedy policy with high, gradually decaying $\epsilon$, leading to substantial action diversity and many suboptimal exploratory actions. Although the behavioral policies used to generate this agent are not especially competent (see Table 1), they have above-random performance on almost all games, suggesting that that this dataset includes more task-relevant transitions.

- **(M)ixed** Finally, for a realistic best-case scenario, we create a dataset of both medium and low-quality data. To simulate a real-world collection of data from different policies, we concatenate multiple checkpoints evenly spread throughout training of a DQN. We believe this is also a reasonable approximation for data from a modern unsupervised exploration method such as CPT (Campos et al., 2021); as shown in Table 1, the agent for this dataset has performance in between CPT and VISR, with median closer to CPT and mean closer to VISR. This data is also lower quality than the expert data originally used in the method most similar to ours, ATC (Stooke et al., 2021). [3] We create a dataset of 3M steps and a larger dataset of 6M steps; all **M** experiments use the 3M step dataset unless otherwise noted.

We compare the agents used for our datasets to those for unsupervised exploration pretraining baselines in Table 1. We estimate the performance of the Weak and Mixed agents as the average of the corresponding logged evaluations in the Dopamine (Castro et al., 2018) baselines. Even our largest dataset is quite small compared to the amounts of data gathered by unsupervised exploration methods (see the "Data" column in Table 1); this is intentional, as we believe that unsupervised interactional data may be expensive in real world applications. We show the performance of the non-random data collection policies in Table 2 (note that a fully-random policy has a score of **0** by definition).

Table 1: Performance of agents used in pretraining data collection compared to external baselines on 26 Atari games (Kaiser et al., 2019)

| Method | Mdn | Mean | >H | >0 | Data |
|---|---|---|---|---|---|
| *Exploratory Pretraining Baselines* | | | | | |
| VISR@0 | 0.056 | 0.817 | 5 | 19 | 250M |
| APT@0[1] | 0.038 | 0.476 | 2 | 18 | 250M |
| CPT@0 | **0.809** | **4.945** | **12** | 25 | 4B |
| *Offline Datasets* | | | | | |
| Exploratory | 0.039 | 0.042 | 0 | 18 | 6M |
| Weak[2] | 0.028 | 0.056 | 0 | 23 | 5M |
| Mixed[2] | 0.618 | 1.266 | 10 | **26** | 3M |

[1] Calculated from ICLR 2021 OpenReview submission; unreported in arXiv version.
[2] Upper-bound estimate from averaging evaluation performance of corresponding agents in Dopamine.

### 4.3 Training Details

We optimize our three representation learning objectives jointly during unsupervised pretraining, summing their losses. During finetuning, we optimize only the reinforcement learning and forward dynamics losses, following Schwarzer et al. (2021) (see Section 5.5), and lower the learning rates for the pretrained encoder and dynamics model by two orders of magnitude (see Section 5.4).

We consider the standard three-layer convolutional encoder introduced by Mnih et al. (2015), a ResNet inspired by Espeholt et al. (2018), as well as an enlarged ResNet of the same design. In other respects, our implementation matches that of SPR and is based on its publicly-released code. Full implementation and hyperparameter details are provided in Appendix C. We refer to agents by the model architecture and pretraining dataset type used: **SGI-R** is pretrained on Random, **SGI-E** on Exploratory, **SGI-W** on Weak, and **SGI-M** on Mixed. To investigate scaling, we vary the size of the encoder used in **SGI-M**: the standard Mnih et al. (2015) encoder is **SGI-M/S** (for small), our standard ResNet is simply **SGI-M** and using a larger ResNet is **SGI-M/L** (for large)[4]. For **SGI-M/L** we also use the 6M step dataset described earlier. All ablations are conducted in comparison to **SGI-M** unless

---

[3]Our data-collection agents are weaker than those used by ATC on seven of the eight games they consider.
[4]See Appendix C for details on these networks

otherwise noted. Finally, agents without pretraining are denoted **SGI-None**; SGI-None/S would be roughly equivalent to SPR (Schwarzer et al., 2021).

For baselines, we compare to no-pretraining Atari 100k methods (Kaiser et al., 2019; van Hasselt et al., 2019; Kostrikov et al., 2021; Schwarzer et al., 2021). For our models trained on Random and Exploratory data we compare against previous pretraining-via-exploration approaches applied to Atari 100k (Liu & Abbeel, 2021; Hansen et al., 2020; Campos et al., 2021). In the higher quality data regime, we compare to recent work on data-agnostic unsupervised pretraining, ATC (Stooke et al., 2021), as well as behavioural cloning (BC).

## 5 Results and Discussion

We find that SGI performs competitively on the Atari-100K benchmark; presenting aggregate results in Table 2, and full per-game data in Appendix E. Our best setting, **SGI-M/L**, achieves a median HNS of 0.753, approaching human-level sample-efficiency and outperforming all comparable methods except the recently proposed CPT (Campos et al., 2021). With less data and a smaller model, **SGI-M** achieves a median HNS of 0.679, significantly outperforming the prior method ATC on the same data (**ATC-M**). Meanwhile, **SGI-E** achieves a median HNS of 0.456, matching or exceeding other exploratory methods such as APT (Liu & Abbeel, 2021) and VISR (Hansen et al., 2020), as well as ATC-E.

**Pretraining data efficiency** SGI achieves strong performance with only limited pretraining data; our largest dataset contains 6M transitions, or roughly 4.5 days of experience. This compares favourably to recent works on unsupervised exploration such as APT or CPT, which require far larger amounts of data and environment interaction (250M steps or 193 days for APT, 4B steps or 8.45 years for CPT). We expect SGI would perform even better if used in these large-data settingss, as we find that it scales robustly with data (see Section 5.2).

**Behavioural cloning is a strong baseline** Although ATC pretrains with expert data, they did not investigate behavioral cloning as a baseline pretraining objective. We do so on our **Mixed** dataset, the only one to be generated by policies with significantly above-random performance. Behavioral cloning without finetuning (**BC-M@0**) performs poorly, perhaps due to the varying behavioural quality in the dataset. But when finetuned, **BC-M** yields very respectable performance, surpassing **ATC-M** but not **SGI-M**. All fine-tuning settings for BC-M match SGI-M.

### 5.1 Data quality matters

In principle, SGI can be used with any offline dataset but we demonstrate that it scales with

Table 2: HNS on Atari100k for SGI and baselines.

| Method | Mdn | Mn | >H | >0 | Data |
|---|---|---|---|---|---|
| *No Pretraining (Finetuning Only)* | | | | | |
| SimPLe | 0.144 | 0.443 | 2 | **26** | 0 |
| DER | 0.161 | 0.285 | 2 | **26** | 0 |
| DrQ | 0.268 | 0.357 | 2 | 24 | 0 |
| SPR | 0.415 | 0.704 | 7 | **26** | 0 |
| SGI-None | 0.343 | 0.565 | 3 | 26 | 0 |
| *Exploratory Pretraining + Finetuning* | | | | | |
| **Method** | **Mdn** | **Mn** | **>H** | **>0** | **Data** |
| VISR | 0.095 | 1.281 | 7 | 21 | 250M |
| APT | 0.475 | 0.666[1] | 7 | 26 | 250M |
| CPT@0[2] | **0.809** | **4.945** | 12 | 25 | 4000M |
| ATC-R[3] | 0.191 | 0.472 | 4 | **26** | 6M |
| ATC-E[3] | 0.237 | 0.462 | 3 | **26** | 6M |
| SGI-R | 0.326 | 0.888 | 5 | **26** | 6M |
| SGI-E | 0.456 | 0.838 | 6 | **26** | 6M |
| *Offline-data Pretraining + Finetuning* | | | | | |
| **Method** | **Mdn** | **Mn** | **>H** | **>0** | **Data** |
| ATC-W[3] | 0.219 | 0.587 | 4 | **26** | 3M |
| ATC-M[3] | 0.204 | 0.780 | 5 | **26** | 3M |
| BC-M@0 | 0.139 | 0.227 | 0 | 23 | 3M |
| BC-M | 0.548 | 0.858 | 8 | **26** | 3M |
| SGI-W | 0.589 | 1.144 | 8 | **26** | 5M |
| SGI-M/S | 0.423 | 0.914 | 8 | **26** | 3M |
| SGI-M | 0.679 | 1.149 | **9** | **26** | 3M |
| SGI-M/L | **0.753** | **1.598** | **9** | **26** | 6M |

[1] APT claims 0.6955, but we calculate 0.666 from their reported per-game scores.
[2] CPT@0 does not do any finetuning.
[3] Our implementation (see Appendix C.6)

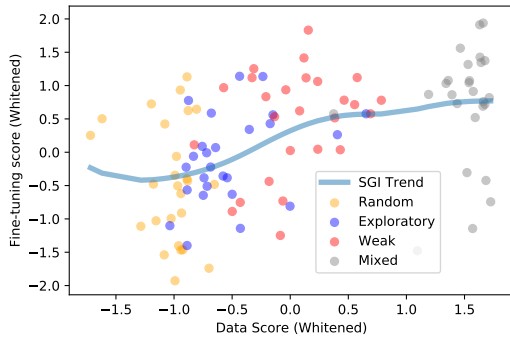

Figure 2: SGI finetuning performance vs. pretraining data score for all combinations of game and dataset. Data score is estimated as clipped return per episode, trend calculated via kernel regression. Values whitened per-game for clarity.

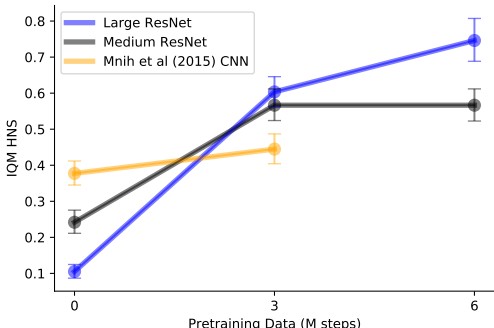
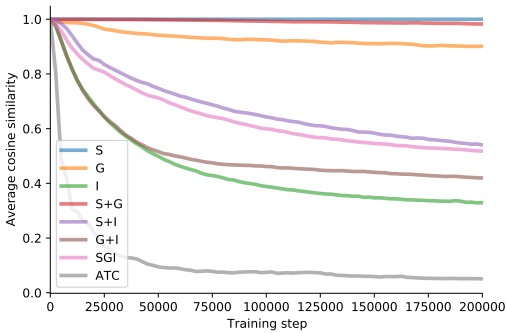

Figure 3: Finetuning performance of SGI for different CNN sizes and amounts of pretrained data from the **Mixed** dataset. We plot IQM HNS with confidence intervals (see Appendix B).

Figure 4: Average cosine similarity between representations over pretraining, averaged across the 26 Atari 100k games. 1 indicates representations are identical, 0 perfect dissimilarity.

the quality of its data. Near the lower bound of data quality where all actions are selected randomly, **SGI-R** still provides some benefit over an otherwise-identical randomly-initialized agent (**SGI-None**) on 16 out of 26 games, with a similar median but 57% higher mean HNS. With better data from an exploratory policy, **SGI-E** improves on 16/26 games, gets 33% higher median HNS, and surpasses APT (Liu & Abbeel, 2021) which used 40 times more pretraining data. With similarly weak data but possibly more task-specific transitions, **SGI-W** gets 72% higher median HNS compared to SGI-None and with realistic data from a mixture of policies, **SGI-M** improves to 98%.

Importantly, the pattern we observe is very different from what would be expected for imitation learning. In particular, SGI-W's strong performance shows that expert data is not required. To characterize this, we plot the average clipped reward [5] experienced per episode for each of our pretraining datasets in Figure 2. Normalizing across tasks, we find a strong positive correlation between task reward engagement ($p < 0.0001$) and finetuning performance. Moreover, we find diminishing returns to further task engagement.

### 5.2 Pretraining unlocks the value of larger networks

The three-layer network introduced by Mnih et al. (2015) has become a fixture of deep reinforcement learning, and has been employed by previous works examining pretraining in this field (e.g. Liu & Abbeel, 2021; Stooke et al., 2021). However, we find that representational pretraining with this network (**SGI-M/S**) provides only minor benefits compared to training from scratch. In contrast, larger networks struggle without pretraining but shine when pretrained as shown in Figure 3.

This finding is consistent with recent work in unsupervised representation learning for classification, which has observed that unsupervised pretraining benefits disproportionately from larger networks (Chen et al., 2020a). In particular, our results suggest that model size should increase in parallel with amount of pretraining data, matching recent work on scaling in language modeling (Kaplan et al., 2020; Hernandez et al., 2021). SGI thus provides a simple way to use unlabeled data to extend the benefits of large networks, already well-known in the large-data regime (e.g., Schrittwieser et al., 2021; Espeholt et al., 2018), to data-efficient RL.

### 5.3 Combining SGI's objectives improves performance

We test all possible combinations of our three SSL objectives, denoted by combinations of the letters S, G and I to indicate which objectives they employ. Results in Table 3 show that performance monotonically increases as more objectives are used, with inverse dynamics modeling combined with either of the other objectives performing respectably well. This illustrates the importance of using multiple objectives to obtain representational coverage of the MDP.

---

[5]Unclipped rewards are not available for the offline DQN dataset.

We note that including inverse modeling appears to be critical, and hypothesize that this is related to representational collapse. To measure this, we plot the average cosine similarity between representations $y_t$ of different states for several pretraining methods in Figure 4, using our ResNet encoder on the Mixed dataset. We observe that **S**, **G** and **S+G** all show some degree representational collapse, while configurations that include inverse dynamics modeling avoid representational collapse, as does ATC, whose contrastive loss implicitly optimizes for representational diversity (Wang & Isola, 2020). Intriguingly, we observe that increased representational diversity does not necessarily improve performance. For example, SGI outperforms **ATC**, **G+I** and

Table 3: HNS on Atari 100K for pretraining ablations of SGI.

| Pretraining | Mdn | Mean | >H |
|---|---|---|---|
| None | 0.343 | 0.565 | 3 |
| S | 0.009 | -0.054 | 0 |
| G | 0.060 | 0.181 | 1 |
| I | 0.411 | 0.943 | 7 |
| S+G | 0.029 | 0.098 | 0 |
| G+I | 0.512 | 1.004 | **9** |
| S+I | 0.629 | 0.978 | 8 |
| SGI-M | **0.679** | **1.149** | **9** |

**I** in finetuning but has less diverse pretraining representations. We also observe that adding SPR (**S**) consistently pulls representations towards collapse (compare **S+I** and **I**, **S+G** and **G**, and **SGI** and **G+I**); how this relates to performance is a question for future work.

## 5.4 Naively finetuning ruins pretrained representations

We find that properly finetuning pretrained representations is critical, as results in Table 4 show. Although allowing pretrained weights to change freely during finetuning is better than initializing from scratch (**Naive** vs **No Pretrain**), freezing the pretrained encoder (**Frozen**) leads to better performance than either. SGI's approach of reducing finetuning learning rates for pretrained parameters leads to superior performance (**Reduced LRs**, equivalent to **SGI-M**).

Table 4: HNS on Atari 100K for finetuning schemes for SGI.

| Method | Mdn | Mean | >H |
|---|---|---|---|
| No pretrain | 0.343 | 0.565 | 3 |
| Naive | 0.429 | 0.845 | 8 |
| Frozen | 0.499 | 0.971 | 8 |
| Reduced LRs | **0.679** | **1.149** | **9** |

We thus hypothesize that representations learned by SGI are being disrupted by gradients early in finetuning, in a phenomenon analogous to catastrophic forgetting (Zenke et al., 2017; Hadsell et al., 2020). As representations may not generalize between different value functions across training (Dabney et al., 2021), allowing the encoder to strongly adapt early in training could make it *worse* at modeling later value functions, compared to the neutral initialization from SGI. We also note that there is a long history in computer vision of employing specialized finetuning hyperparameters (Li et al., 2020; Chu et al., 2016) when transferring tasks.

## 5.5 Not all SSL objectives are beneficial during finetuning

Although SGI uses **S** during finetuning, we experiment with a variant that optimizes only the standard DQN objective, roughly equivalent to using DrQ (Kostrikov et al., 2021) with DQN hyperparameters set to match SGI. We find that finetuning with **S** has a large impact with or without pretraining (compare **None** and **S Only** entries in Table 5.). Although, SGI without **S** manages to achieve roughly the same mean human-normalized score as SGI with **S**, removing **S** harms performance on performance on 19 out of 26 games and reduces median normalized score by roughly 33%. We also find no benefit to using all of SGI's constituent objectives during finetuning (**All Losses** in Table 5) compared to using **S** alone, although the gap between them is not statistically significant for metrics other than median (see Figure 5d).

Table 5: HNS on Atari 100K for finetuning ablations of SGI.

| Finetune SSL | Mdn | Mean | >H |
|---|---|---|---|
| *Without SGI pretraining* | | | |
| None | 0.161 | 0.315 | 2 |
| S Only | 0.343 | 0.565 | 3 |
| *With SGI pretraining* | | | |
| None | 0.452 | 1.114 | 8 |
| SGI | 0.397 | 1.011 | 8 |
| S Only | **0.679** | **1.149** | **9** |

# 6 Conclusion

We present SGI, a fully self-supervised (reward-free) approach to representation learning for reinforcement learning, which uses a combination of pretraining objectives to encourage the agent to learn multiple aspects of environment dynamics. We demonstrate that SGI enables significant improvements on the Atari 100k data-efficiency benchmark, especially in comparison to unsupervised exploration approaches which require orders of magnitude more pretraining data. Investigating the various components of SGI, we find that it scales robustly with higher-quality pretraining data and larger models, that all three of the self-supervised objectives contribute to the success of this approach, and that careful reduction of fine-tuning learning rates is critical to optimal performance.

# 7 Acknowledgements

We would like to thank Mila and Compute Canada for computational resources. We would like to thank Rishabh Agarwal for useful discussions. Aaron Courville would like to acknowledge financial support from Microsoft Research and Hitachi. We would also like to thank Wancong (Kevin) Zhang for his helpful comments to the codebase on GitHub.

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
