# OpenReview forum: "Pretraining Representations for Data-Efficient Reinforcement Learning"
_NeurIPS.cc/2021/Conference — NeurIPS 2021 Poster_

### Official Review · Reviewer_5yeQ · 2021-07-11

**Rating:** 7
**Confidence:** 4

**Summary:**

The proposed method combines 3 unsupervised representation learning objectives (latent dynamics modelling, goal-conditioned RL, and inverse dynamics modeling) in order to pre-train representations for RL. The experiments on the Atari 100K benchmark show that SGI improves performance over previous representation pretraining baselines (ATC, CPT), and is competitive with exploration-based approaches (APT, VISR) which require more pretraining data and interactions with the environment.

Summary of experimental results:

(1) Data-efficiency: Table1 shows that SGI achieves comparable performance as APT and CPT (unsupervised exploration methods) but using far fewer data and environment interactions (250M steps for APT, 4B steps for CPT, 6M for SGI).

(2) Varying the quality of the offline pre-training data (collected by a random policy, SGI-R; an exploratory policy, SGI-E; suboptimal DQN actions, SGI-W; and DQN agent at different checkpoints, SGI-M): SGI’s performance increases as data comes from better-performing or more-exploratory policies.

(3) Varying the size of the encoder used in SGI-M: SGI-M/S (small), SGI-M/L (large): Larger model size results in increased performance.

(4) Pretraining vs. size of the network (Section 5.2): Model size and amount of pretraining data should increase in parallel.

(5) Ablation study (Section 5.3, Table 3): Inverse dynamics modeling objective seems to be the most critical to performance. Performance increases as more objectives are used.

(6) How to finetune pre-trained representations properly: Reducing the finetuning LRs improves performance.


**Limitations And Societal Impact:**

Yes

**Main Review:**

Originality & Significance: The largest contribution of the paper are the extensive experiments and empirical analyses (Section 4), which I believe are valuable for understanding how pre-trained representations can help RL with learning efficiency. (See “Summary” above for a list of empirical findings reported in the paper.)

The proposed method itself (SGI) is less novel -- it is a combination of three unsupervised representation learning objectives that operate in latent space provided by a shared encoder. However, the ablation analysis (Section 5.3) is helpful in understanding how the combination of the three objectives contributes to SGI’s performance.

Quality: The experiments are thoughtfully designed and well-explained (evaluation metrics, datasets, empirical analyses and discussion of results).

Clarity: The paper is very well-written; the problem motivation, experimental design, related works, evaluation metrics, etc. are all clearly explained.

Minor nit suggestions that did not affect my score:
- L71: Missing question mark “?”
- L293: “settingss” -> “settings”


**Time Spent Reviewing:**

3

---

> ### Author Response · Authors · 2021-08-10
> **Author response to reviewer 5yeQ**
>
> Thank you for your review!  We’re glad you appreciated our work, and we will address your suggestions in the next version.
>
> Regarding SGI’s novelty, we agree that SGI is a combination of known methods.  We see this as a demonstration of the benefits of revisiting and synthesizing existing methods from a fresh perspective.  For example, goal-conditioned reinforcement learning has to the best of our knowledge not been used as a representation learning task in prior work, even though goal-conditioned reinforcement learning itself is very widespread.

---

### Official Review · Reviewer_2aRe · 2021-07-14

**Rating:** 6
**Confidence:** 4

**Summary:**

This paper proposes a new approach for pretraining an encoder using unlabeled data which is then used for a downstream RL task, leading to large improvements in sample efficiency. The approach is based on latent dynamics modeling and unsupervised goal-conditioned RL. The approach shows significant gains in performance and sample efficiency on Atari 100k and scales well with model size and better data.

**Main Review:**

This paper introduces a rather simple yet effective approach for learning good representations in an unsupervised way which can be helpful for downstream RL tasks. This problem is very important as it can lead to more sample efficient algorithms and thus enable more real-world applications of RL where interactions with an environment are expensive. The paper is clear overall, the experiments are well designed, and the approach is thoroughly evaluated and compared with many relevant baselines and ablations. The results also look quite promising and the method clearly improves performance and sample efficiency relative to prior art.

I particularly liked the analysis of the algorithm's dependence on the quality of the dataset and found the insights very valuable even if not necessarily surprising. The paper also contains many ablations thus resulting in a better understanding of how each component contributes to the final results.

However, I think there are a few additional baseline variants / ablations which could further improve the paper and our understanding of how the various algorithms compare. Would it be possible to add experiments with BC, VISR, APT, CPT, ATC, SGI using the all of the four different datasets considered, namely R, E, W, and M? Some of these are already included in Table 2 but completing this space would be useful to gain a full picture. For Table 5, could you add results with G only and I only?

From Table 4, it looks like Frozen is comparable with BC-M so it seems that reducing the learning rate during finetuning is quite important to maximize performance. Hence, it would be interesting to see the effect of reducing the learning rate on the other baselines which perform finetuning on downstream tasks. Is the right conclusion that reducing the learning rate during finetuning plays a more important role than the objective function used for pretraining the encoder? I think further analyzing this question would be of interest for the readers.

I think it would also be valuable to add experiments in a different domain in order to better understand whether the conclusions drawn from the Atari experiments generalize to other domains as well. A few suggestions are evaluating the methods on continuous control tasks from MuJoCo or DMC.

Given the above reasons, I tend to recommend acceptance of this paper, but encourage the authors to take into account the feedback to further improve this paper.

**Time Spent Reviewing:**

5 hours

---

> ### Author Response · Authors · 2021-08-10
> **Author response to reviewer 2aRe**
>
> Thank you for your thoughtful review!  We address your comments in detail below.
>
> > “However, I think there are a few additional baseline variants / ablations which could further improve the paper and our understanding of how the various algorithms compare. Would it be possible to add experiments with BC, VISR, APT, CPT, ATC, SGI using the all of the four different datasets considered, namely R, E, W, and M? Some of these are already included in Table 2 but completing this space would be useful to gain a full picture”
>
> Thanks, this is a great observation, and yes, controlling for the dataset itself would be valuable. However, it isn't possible to present APT/CPT/VISR results on the other datasets, as all of these algorithms are exploration methods that dynamically generate their own dataset during training -- unlike SGI, which uses a fixed dataset and does not require extra environment interactions.  While in principle one might be able to run these algorithms with offline data like SGI, they were not designed for that setting and might not perform well.  We believe that SGI’s ability to use a fixed dataset is a distinct advantage, and will update the text to highlight the different experimental settings used in prior work.
>
> That said, there is one other set of experiments that we wish we were able to add: using the datasets generated by these algorithms for SGI.  Since all of these algorithms are better at exploration than the one used to create SGI-E's dataset, we would expect SGI to perform better given their data.  Unfortunately, none of these methods made code or data available, and given their massive computational requirements it would not be practical for us to attempt a replication from scratch.
>
> > From Table 4, it looks like Frozen is comparable with BC-M so it seems that reducing the learning rate during finetuning is quite important to maximize performance. Hence, it would be interesting to see the effect of reducing the learning rate on the other baselines which perform finetuning on downstream tasks. Is the right conclusion that reducing the learning rate during finetuning plays a more important role than the objective function used for pretraining the encoder? I think further analyzing this question would be of interest for the readers.
>
> All of our baselines use identical fine-tuning settings to SGI (with the exception of methods that do not pretrain certain parameter groups; in those cases, non-pretrained parameters are given standard learning rates).  This allows us to rule out the hypothesis that only SGI's fine-tuning settings separate it from previous methods such as ATC.  However, we can add an experiment showing the impact of the modified fine-tuning settings for some of our stronger baselines, such as BC or S+I/G+I.
>
> > “For Table 5, could you add results with G only and I only?”
>
> Yes, this is straightforward to do.

---

> > ### Comment · Reviewer_2aRe · 2021-08-29
> > **Thank you for the useful response**
> >
> > I thank the authors for their response and acknowledge that most of my concerns have been addressed. While I still think running experiments on a different domain would strengthen the paper, I believe the empirical study is thorough enough in its current form to be a valuable contribution at NeurIPS. I agree with reviewer KfDr that the authors should make the claims more specific and narrow in scope in order to reflect the evidence presented in the paper.
> >
> > Assuming the authors will keep their promise to make these changes, I recommend acceptance and look forward to reading the final draft.

---

### Official Review · Reviewer_KfDr · 2021-07-16

**Rating:** 5
**Confidence:** 4

**Summary:**

This paper proposes a combination of three existing self-supervised objectives (self predictive representations, goal conditioned rl and inverse dynamics modeling) to improve the performance of RL agents, after pre-training on unsupervised data and then fine tuning on a small supervised (i.e. task specific) dataset. The authors analyzed the significance of each one of these objectives as well as the importance of the number of parameters and the impact of pre-training data quality.

**Limitations And Societal Impact:**

Minimal description of the limitation and and social impact.

**Main Review:**

==== Writing

The paper is clearly written and is easy to read and understand.  The paper does not introduce any new model or objective but rather analysis a combination of already existing objectives for pre-training and the effect of each one on the performance of the agent on the downstream task. Due to the high number of possible combinations of self-supervised objectives, I really enjoyed the notations used by the paper to make it accessible and easy to read and understand. However, these notations and abbreviations can become overwhelming and I suggest the authors to include an additional summary table which describes them all in one glance.

==== Methodology

In short, the paper is an empirical study of the importance of the pre-training objectives on the performance of the rl agent. And such studies are hard to do by nature because the number of possible combinations can grow exponentially, making experimentations harder. The authors, did a good job keeping the number of experiments at a (relatively) low number by fixing many of variables e.g. the rl algorithm itself which is fixed to Q-learning and the environment which is fixed at Atari.

At the same time, this is a major shortcoming of the paper. For example, one may argue that the gains in Table 2 vs SimPLe with no pre-training can be described by using a fundamentally more efficient rl method (Q-learning vs PPO) just because the paper does not include any study to demonstrate the same performance gains for PPO. Or ATC vs SGI comparison can be "explained away" by insignificance of augmentation for Atari. Or one may argue that SGI is observing better performances because learning a representation of Atari games is easy and SGI will perform the same when the environment is hard to compress (at least naively) such as the The Distracting Control Suite. To summarize, I believe the paper suffers from overfitting on its fixed assets e.g. environment (Atari) and rl method (Q), and in lack of any experiment to cover other cases, this is hard to support or oppose. Ofcourse, it is impossible to run all possible variations but a selective few can substantially support the main claims of the paper. At the same time, the claims can be rewritten in a less generalize form to limit it on the scope of the experiments.

Another major issue of the paper is how it represents the representation learning (no pun intended). The paper claims "stronger" representations however this claim is only support by demonstrating stronger performance on a single downstream task (this is another reason why Atari is not a good environment for this study vs multi-task environments). I encourage the authors to use other representation learning metrics such as Mutual I(z;x) or Linear Regression of reward from z or the number of latent active units.

Overall, I find a disparity between the scope of the claims and the provided empirical evidence which can be improved by toning down the claims and adding more empirical evidence.



**Time Spent Reviewing:**

4

---

> ### Author Response · Authors · 2021-08-10
> **Author response to reviewer KfDr**
>
> Thank you for your in-depth review. As a high-level response to your comments -- it wasn’t our intention to make very general claims around SGI being applicable to any domain (such as distracting DM Control), and we would be happy to review and edit our framing and tone to address this. As you point out, this is an analytical work focused on Atari-100K, which we believe to be the best benchmark at present for measuring data-efficiency in deep RL. We address your specific concerns in detail below.
>
> > “I suggest the authors to include an additional summary table which describes them all in one glance.”
>
> We will add such a table to our updated text.
>
> > “For example, one may argue that the gains in Table 2 vs SimPLe with no pre-training can be described by using a fundamentally more efficient rl method (Q-learning vs PPO) just because the paper does not include any study to demonstrate the same performance gains for PPO.”
>
> We include SimPLe only for historical reasons, as it introduced the Atari 100k task.  It is not meant to be a controlled comparison; it is certainly true that some part of our gains relative to SimPLe are explained by DQNs being more efficient (as can be seen by the fact that SGI-0, our no-pretraining control, substantially outperforms SimPLe), but one could compare to other baseline methods such as DrQ (Kostrikov et al, 2021) and SPR in this regard.
>
> To minimize the odds that any discrepancies (such as in augmentation or Q-learning algorithm) skew our results, as many of our comparisons as possible have been made fully-controlled with matching architectural and hyperparameter settings.  For ATC, we use publicly-released code to ensure matching augmentation and fine-tuning hyperparameters, while all controls implemented in our codebase fully share hyperparameters by default.  Thus, the only differences lie in pretraining, the subject we seek to investigate.
>
> The only comparisons for which this is not the case are between SGI and APT, VISR, and CPT.    These algorithms are themselves not directly comparable to SGI as they require exploratory interaction with the environment during pretraining, and do not learn from offline data. Furthermore, they do not have publicly available code, and due to their computational requirements these algorithms are essentially impossible for us to replicate from scratch, which makes conducting closely-controlled comparisons difficult.
>
> > “one may argue that SGI is observing better performances because learning a representation of Atari games is easy and SGI will perform the same when the environment is hard to compress (at least naively) such as the The Distracting Control Suite.”
>
> Although we cannot know what results SGI might attain on other tasks (and will say so in the text), we do not think learning performant representations for RL on Atari is necessarily “easy” or as simple as compressing the input.  Anand et al (2019) found that learning task-agnostic representations for Atari was non-trivial, with many baselines failing to capture relevant state information. Likewise, many reasonable pretraining methods, such as ATC, do not appear to improve reinforcement learning sample efficiency by large margins.
>
> > “Another major issue of the paper is how it represents the representation learning (no pun intended). The paper claims "stronger" representations however this claim is only support by demonstrating stronger performance on a single downstream task”
>
> We have made a conscious decision to focus on sample efficiency (as defined by downstream performance), rather than learning task-agnostic representations, and will change the language in the paper to specifically state this. Task-agnostic representations for Atari have been investigated by prior work: for example, Anand et al (2019) introduced the contrastive algorithm ST-DIM and used linear probing to evaluate how much information their representations contained about game state, but did not evaluate performance on reinforcement learning directly.  However, ATC claimed better reinforcement learning performance than ST-DIM in a controlled comparison, and SGI far outperforms ATC, suggesting that algorithms like ST-DIM that have been tuned for task-agnostic learning may not be optimal for data-efficient task-specific learning. This is what we meant our representations to be “stronger” for, and we will clarify this.
>
> References:
>
> Anand, A., Racah, E., Ozair, S., Bengio, Y., Côté, M.-A., and Hjelm, R. D. Unsupervised state representation learning in atari. In NeurIPS, 2019.
>
> Kostrikov, I., Yarats, D. and Fergus, R., 2020. Image augmentation is all you need: Regularizing deep reinforcement learning from pixels. In ICLR, 2021

---

### Official Review · Reviewer_Tq2R · 2021-07-16

**Rating:** 7
**Confidence:** 4

**Summary:**

Focusing on the performance of an RL agent given a finite number of samples from ATARI in the regime where it is able to pretrain on a offline dataset, the paper shows that the final performance of the agent is improved by the addition of several additional auxiliary losses during pretraining, along with adjustments to the finetuning step.

**Limitations And Societal Impact:**

Yes, the societal impacts relevant to the paper are tangential and I believe the authors sufficiently describe them.

**Main Review:**

Broadly speaking I believe this paper makes a strong contribution by showing that the proposed techniques improve the finetuned performance of the model, bringing it to substantially higher performance relative to other models in similar regimes.

The evaluation of the models is detailed, the ablations and variations that are considered are reasonable and executed well, and the presentation is good. I believe that work like this which clearly sets out the benefit (and weaknesses) of pretraining approaches is worthy of acceptance. For that reason I am giving a recommendation of 7.

However I have one significant comment, and a set of more minor ones.

1) A substantial part of the improvement in mean/median scores on Atari, as shown in Table 4, comes from the particular finetuning strategy employed (specifically, the reduced LR setting). Without that trick, the performance of the network drops substantially, roughly equivalent to removing all but the Inverse Modeling head from the finetuning tasks (comparing to Table 3), or reverting back to the behavioral cloning baseline (comparing to Table 2). This has substantial impact, and while it is discussed in Section 5.4, I believe it should be give a substantially more central role to the paper, including asking some further questions to probe exactly the cause of the effect (for instance, is performance recovered with a frozen encoder if you make the Q-Learning network deeper?).

Additionally, I have a set of more minor comments:

* [Line 73] You're claiming these questions are policy-indepenent, but isn't the inverse modelling question (line 70) policy-dependent for non-trivial environments? Similarly for the question in line 71, it seems like that is policy-dependent as well.

* [Line 111] It seems worthwhile to me to cite some earlier examples of inverse dynamics modeling here.

* [Line 113] re: "it is naturally not prone to representational collapse". That seems like a broad claim which depends a lot on the situation, can you cite or explain?

* [Figure 3] This plot strikes me as begging to be extended both further to the right and in model size. Especially because you bring up the comparison to [Kaplan, 2020], it would be interesting to see these points (especially if they were extended) plotted as log-logs to try to see whether there are visible patterns in the scaling properties.

* [Line 389] You refer to an All Losses row in Table 5, but no such row exists.

* [Line 391] nit: Can you make clear that Figure 5d is in the appendix?


Finally, I would like to note that while I am familiar with the subject space, I am not fully up to date on recent developments in this area and am not a strong judgement of novelty past the references given in the paper (relative to which I think this submission is sufficiently novel). I will take a look at the other reviews to ensure that I didn't miss anything in my survey.

**Time Spent Reviewing:**

5

---

> ### Author Response · Authors · 2021-08-10
> **Author response to reviewer Tq2R**
>
> Thank you for your review and favourable assessment of our work. We provide a response to your comments below:
>
> > “A substantial part of the improvement in mean/median scores on Atari, as shown in Table 4, comes from the particular finetuning strategy employed (specifically, the reduced LR setting). Without that trick, the performance of the network drops substantially, roughly equivalent to removing all but the Inverse Modeling head from the finetuning tasks (comparing to Table 3), or reverting back to the behavioral cloning baseline (comparing to Table 2). This has substantial impact, and while it is discussed in Section 5.4, I believe it should be give a substantially more central role to the paper, including asking some further questions to probe exactly the cause of the effect (for instance, is performance recovered with a frozen encoder if you make the Q-Learning network deeper?).”
>
> It's definitely true that fine-tuning is important, and we will give it a more prominent position in the final version of the paper. Although a systematic study of how best to finetune pretrained representations is beyond the scope of this paper, we are considering a handful of new experiments to run. As you suggested, one option would be to make the MLP used by the value head deeper while keeping representations frozen; another would be to decay weights to their initialized value (essentially, using a variant of Adam-W where weights decay to an initialization rather than 0); and a third would be to freeze the encoder early in training but finetune it normally on. We will also reorder text and emphasize the importance of this design choice, as well as drawing parallels to finetuning methods used in NLP and CV, where learning rate reduction at finetuning is widespread.
>
> > [Line 73] You're claiming these questions are policy-indepenent, but isn't the inverse modelling question (line 70) policy-dependent for non-trivial environments? Similarly for the question in line 71, it seems like that is policy-dependent as well.
>
> You're correct that all of our tasks are policy-dependent in the sense that how they are optimized will be heavily influenced by the policy of the offline data.  The key idea we wish to communicate by the term is that the tasks can be meaningfully optimized even with data from a fully-random policy (unlike behavioral cloning or other imitation learning); we will search for a better way to express that.
>
> > [Line 113] re: "it is naturally not prone to representational collapse". That seems like a broad claim which depends a lot on the situation, can you cite or explain?
>
> If representations were to collapse, our inverse model head would only be able to predict a fixed distribution p(a), and would not be able to model the conditional structure in p(a|s,s').  In a setting where the transition function depends on the agent's actions (most RL environments), p(a) will not be a good approximation for p(a|s,s').  Thus, representational collapse would lead to a high value of the inverse modeling loss, so optimization will naturally steer representations away from it.  This is the opposite of the case for SPR (or other tasks similar to BYOL, Grill et al, 2020) for which representational collapse actually minimizes the training loss. We will expand on this reasoning in the final version.
>
> > “[Figure 3] This plot strikes me as begging to be extended both further to the right and in model size. Especially because you bring up the comparison to [Kaplan, 2020], it would be interesting to see these points (especially if they were extended) plotted as log-logs to try to see whether there are visible patterns in the scaling properties”
>
> We fully agree that extending the curve to the right would be very interesting.  Unfortunately, we may not have the resources ourselves to scale far beyond our current largest model, but we hope that another group will do so.
>
> > [Line 111] It seems worthwhile to me to cite some earlier examples of inverse dynamics modeling here.
>
> We will do so.
>
> > [Line 389] You refer to an All Losses row in Table 5, but no such row exists.
>
> This was meant to refer to the row currently marked as SGI; we will correct the text.
>
> > [Line 391] nit: Can you make clear that Figure 5d is in the appendix?
>
> We will update the text to address these comments.
>
> References:
> Grill, J.B., Strub, F., Altché, F., Tallec, C., Richemond, P.H., Buchatskaya, E., Doersch, C., Pires, B.A., Guo, Z.D., Azar, M.G. and Piot, B., 2020. Bootstrap your own latent: A new approach to self-supervised learning. In NeurIPS, 2020

---

### Decision · Program_Chairs · 2021-09-27

**Decision:**

Accept (Poster)

**Comment:**

Taking as evaluation setting Atari 100k, the paper demonstrates the effectiveness of pretraining an agent torso/encoder on a collection of auxiliary self supervised losses, greatly improving data efficiency and final performance of the agent when training on a small of amount of experience on the downstream tasks (in conjunction with careful tuning of fine-tuning learning step).

Whilst the proposed unsupervised losses are not novel (a combination of latent dynamics modelling, goal-conditioned RL, and inverse dynamics modeling), the most impactful contribution of the submission is the thorough evaluation of the method, its extensive, carefully designed experiments, ablations and empirical analyses, as well as the solid choice of baselines. Although limited to Atari 100k, these experiments are well described and discussed, forming a very solid foundation for eventual follow-up work investigating if these results can generalize to other domains or training regimes,

I believe the paper will be of great interest to the broader community, as it sheds light on viable training regimes for agents in real-world application, where access to real environment experience is scarce.